# Social information use in adolescents: The impact of adults, peers and household composition

**Lucas Molleman** [1,2,3]*, **Patricia Kanngiesser** [4,5], **Wouter van den Bos**[1,2,3]

**1** Amsterdam Brain and Cognition, University of Amsterdam, The Netherlands, **2** Department of Developmental Psychology, University of Amsterdam, The Netherlands, **3** Center for Adaptive Rationality, Max Planck Institute for Human Development Berlin, Germany, **4** Faculty of Education and Psychology, Freie Universität Berlin, Germany, **5** Faculty of Education, Leipzig University, Germany

* l.s.molleman@uva.nl

**Data Availability Statement:** The data for this article is openly available on GitHub: https://github.com/LucasMolleman/Molleman_Kanngiesser_vandenBos.

## Abstract

Social learning strategies are key for making adaptive decisions, but their ontogeny remains poorly understood. We investigate how social information use depends on its source (adults vs. peer), and how it is shaped by household composition (extended vs. nuclear), a factor known to modulate social development. Using a simple estimation task, we show that social information strongly impacts the behaviour of adolescents aged 11 to 15 years (N = 256), especially when its source is an adult. However, social information use does not depend on household composition: the relative impact of adults and peers was similar in adolescents from both household types. Furthermore, adolescents were found to directly copy others' estimates surprisingly frequently. This study provides novel insights into adolescents' social information use and contributes to understanding the ontogeny of social learning strategies.

## Introduction

Observing others may provide useful information that allows individuals to make better and more accurate decisions, while avoiding the potential costs of trial and error [1–3]. Social learning is considered fundamental for the evolution of cumulative culture, as it allows for the gradual improvement of skills and knowledge as well as their spread through populations [4–8]. Individuals tend to use social information strategically, being selective as to when to learn from others, and whom to learn from [9–11]. Choosing social sources as a 'model' to acquire information from is a key element of many social learning strategies, and is often based on the model's prestige, success, familiarity, or age [3,12–14].

Although there is a wealth of research on social information use in childhood [14–20], systematic studies in adolescence are rare. This paucity of experimental studies is surprising given that adolescents undergo a process of major social re-orientation in which their behaviour becomes increasingly influenced by peers [21–24]. In this paper, we experimentally investigate how social information use in early to middle adolescence is influenced by (*i*) the source of

**Funding:** L.M. and W.v.d.B. were supported by Open Research Area (ID 176); L.M. is further supported by an Amsterdam Brain and Cognition Project grant 2018. W.v.d.B. is further supported by the Jacobs Foundation, European Research Council grant (ERC-2018-StG-803338) and the Netherlands Organization for Scientific Research grant (NWO-VIDI 016.Vidi.185.068). P.K. is supported by a Freigeist Fellowship from Volkswagen Foundation [no. 89611]. The funders had no role in study design, data collection and analysis, decision to publish, or preparation of the manuscript.

**Competing interests:** The authors have declared that no competing interests exist.

information (adults vs. peer), and (*ii*) the environment of development (household composition; nuclear vs. extended).

Children rely heavily on social information to acquire knowledge and skills [17,19,25,26], and typically prefer to learn from adults over peers [14,27] – especially when they consider adults to be more knowledgeable [28–30] (but see [31]). During adolescence, however, this focus on adults tends to shift as conformity to parental advice and the perceived epistemic authority of parents decreases [32,33]. At the same time, adolescents show an increased sensitivity to peer influence [24,34,35], suggesting that peers may become the most important source of social influence during this developmental period. However, evidence from experiments and the field suggests that the relative influence of peers and adults on adolescents' behaviour depends on the domain of decision making. For example, peers can have a stronger effect on adolescents' willingness to adopt social norms [36], while adults may have more impact in matters of objective knowledge about the environment [37].

Recent experiments have documented substantial variation in social information use between individuals and societies [38–43]. Evidence from studies with adults suggests that, like many aspects of social cognition, individuals' social information use is shaped by socio-economic and cultural conditions [18,41,43–46]. Studies with adolescents indicate that their socio-cultural environment can modulate the relative influence of peers and adults on their behaviour [36,47,48]. In this study, we use a controlled behavioural experiment to shed light on ontogenetic dynamics that might underly these differences. Rather than comparing samples across cultural settings (*e.g.*, between different countries)–where participants may systematically differ in a range of aspects–we investigated social information use in adolescents from come from the same socio-cultural environment, but who differ in one aspect known to shape social and emotional development: the composition of the household they are growing up in.

Household composition has been shown to influence child and adolescent development across a range of domains, including educational outcomes, health outcomes, relationship formation, and emotional development [49–54]. Across the globe, two common types of households are 'nuclear' (comprising parents and their offspring) and 'extended' households (also including grandparents). There are reasons to suspect that household composition may influence the relative importance of adults and peers as sources of social information. Compared to nuclear households, extended households consist of more adults, and individuals growing up in these households tend to associate seniority more strongly with authority and power [55]. Recent field evidence suggests that the extent to which people use social information increases with its availability and value in day-to-day interactions [45]. It is therefore conceivable that adolescents from extended households would value information from adults more due to the benefits of heeding adults' advice in their everyday family life.

We experimentally compared social information use in two groups of 11- to 15-year-old adolescents, from either nuclear or extended households. We recruited adolescents from middle-class families in an urban area in India, where about half of the households consist of extended families. Importantly, the two groups were very similar in terms of socio-economic status, parental education, and religious affiliation; they were recruited from the same schools and they attended the same school classes (see Methods for details). In many parts of India, extended families are patrilocal, that is, women join their husband's family after marriage, and authority traditionally resides with the oldest male in the family [55]. Following international trends [56–58], India has seen a decline of the fraction of extended families over the past few years, and nuclear families have become increasingly common, especially in urban areas [49,59,60].

We used a validated incentivised perceptual decision-making task [61] to measure adolescents' basic propensity to use social information (Fig 1). In this task, participants had to

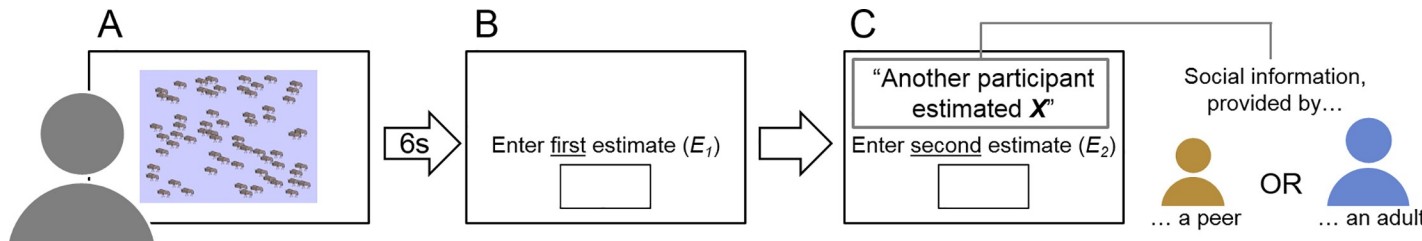

**Fig 1. Experimental task. a,** Participants observed an image showing a group of animals for 6 seconds and had to estimate the number of animals. **b,** Once the image had disappeared, participants entered their first estimate ($E_1$). **c,** Then, they observed social information ($X$; the estimate of another participant who completed the task before without social information) and entered their second estimate ($E_2$). This procedure was repeated for five rounds, in each of two experimental conditions, in which social information was provided by either a peer student (yellow) or an adult (blue). In each round, social information use was calculated as $s = (E_2 - E_1) / (X - E_1)$, and for each experimental condition, each participant's social information use was characterized as the average value of $s$ across the five rounds. Participants did not receive any feedback about their accuracy (nor the accuracy of the social information) between trials.

estimate how many animals were shown in an image, and could adjust their estimate after observing social information. They were shown the estimate of another person who was not known to the participant and who had completed the task before. We used the average degree of participants' estimate adjustment as a measure of their social information use (see Methods for details). In two experimental conditions (within-subject), we varied whether the person who provided the estimate was an unfamiliar peer or an adult.

We had two predictions about the outcome of our experiment. First, we predicted that adolescents would be more influenced by adults than by peers, as the task contained objectively correct solutions [37]. That is, adolescents would adjust their estimates more after observing the estimates of an adult model than after observing the estimates of a peer model. Second, we predicted that the difference between adult and peer models would be particularly pronounced for adolescents from extended households. In other words, we hypothesized that for adolescents from nuclear households the difference between peers and adults would be relatively small, and for adolescents from extended households, this difference would be relatively large.

## Material and methods

### Participants

Across two schools in Pune, India, a total of 264 adolescents (154 from 1 school and 110 from another; age range 11–15 years, mean 13.38, s.d. 0.88; 129 male, 135 female) participated in our tablet-based experiment. In our sample, 148 of the adolescents were from extended households (defined as including a cohabiting grandparental generation), and 108 were from nuclear households. Eight additional adolescents completed the experimental task, but were dropped from the analyses because information on their household composition was not provided.

Pune is a large city with about 4 million inhabitants; and its metropolitan area has a population of more than 5 million. Main industries are IT and manufacturing; the city has a per capita income ranking the sixth highest in India. In Pune, about half of households consist of extended families. The comparison groups in our sample (adolescents from nuclear versus extended households) did not differ with respect to their age and gender distributions, the number of children in their household, or their composition in terms of religious affiliation (Table A in S1 File). In addition to these demographic similarities, we also observe that their parents' educational and occupational backgrounds are similar across the two comparison groups (Table B in S1 File). The adolescents from the two groups in our sample were recruited

from the same schools and attended the same classes. In sum, the context of our study offers an ideal test case to study the effect of household composition on social information use.

## Procedure

Before data collection, the procedures and protocol of this study were approved by the IRB of the Max Planck Institute for Human Development Berlin (approval included data collection in India; IRB approval number ARC 2018/15). The study was conducted by a local researcher in Pune in August and September 2018. Heads of schools gave their approval to conduct the study and handed out consent forms to parents of adolescents in the appropriate age range. Participation was by assent from the adolescents and informed consent from their parents. The consent form was also used to collected the data about household composition (*i.e.*, a list of persons currently living in the same household, including family relationships) as well as parents' educational and professional background. Before starting the task, participants were informed of all procedures, including how their choices affected their earnings. The task was administered on tablet computers and all instructions were in English. Participants attended English middle schools and were proficient in English. The on-screen instructions also included extensive illustrations, test trials and compulsory control questions to ensure that participants fully understood their task (see Supporting Information in S1 File for experimental materials). The local experimenter who, in addition to English, spoke the main local languages (Hindi, Marathi) was always present to answer any questions.

Participants completed an adapted version of the BEAST, a perceptual judgment task that has been shown to reliably measure individuals' use of social information in a limited amount of time [61]. The task consisted of two blocks of 5 rounds each, the order of which was counterbalanced between participants. In one block, participants were informed that they could observe social information from a peer (a student of roughly the same age) model; in the other block, they could observe an adult model. To avoid confounding effects, we did not provide any further information about these models (*e.g.*, regarding their gender, exact age, skill at this task, or other indicators of prestige or success; see Supporting Information for screenshots of the decision screens).

In each round of the task, participants viewed an image containing 50–60 animals (Fig 1A). In each round, we used a different animal species [61]. After 6 seconds, the image disappeared, and participants were asked to estimate how many animals were displayed (Fig 1B). Piloting with local students showed that this range of numbers of animals and the viewing time gave participants a rough impression of the total number of animals but prevented them from actually counting them. The value participants entered after viewing the image was their first estimate in a round (denoted $E_1$). After entering their first estimate, participants were provided with social information (denoted $X$): the estimate of another person who had already completed the task without social information (Fig 1C).

Because the impact of social information on behaviour can vary with its distance from an individual's prior beliefs [42,62–67], we experimentally controlled this distance by selecting social information 15–25% away from an participant's first estimate in a round, using the true value as a reference point. In each round, after a participant had submitted their first estimate, we calculated a 'target' value of social information. We set this target value at $X' = E_1 \cdot (1 + \Delta)$ if the first estimate was lower than correct value, and at $X' = E_1 \cdot (1 - \Delta)$ if the first estimate was higher than the correct value. The value of $X$ displayed to the participant was the value in the pre-recorded sample that was closest to that 'target' value $X'$. When the first estimate was exactly correct, a coin flip determined whether the displayed social information was lower or higher than the first estimate. This procedure ensures that social information was sometimes

lower and sometimes higher than a participant's first estimate. Furthermore, most of the time (but not always), social information was closer to the true value than a participant's first estimate. For rounds 1–5 of the task, we set $\Delta$ to 0.25, 0.15, 0.20, 0.15, and 0.25, respectively.

The pre-recorded sample consisted of 24 adults and 14 students, who completed the task without social information. To obtain a 'saturated' sample of estimates for each round of the actual experiment, the participants in the pre-recorded sample went through 6 iterations of the 5 rounds of the task. As a result, the pre-recorded sample consisted of 144 adult estimates and 84 student estimates for each round to be displayed as social information.

After observing the social information, participants made a second estimate ($E_2$; Fig 1C). For each round, we calculate the relative distance a participant moved towards the social information as $s = (E_2 - E_1) / (X - E_1)$. We define an individual's use of social information as the mean value of $s$ across 5 rounds (denoted $S$). Here, we characterise an individual with two measures: $S_{peer}$ and $S_{adult}$ (to reflect responses to social information provided by a peer student or an adult, respectively). The value of $S_i$ can be viewed as the relative weight an individual tends to assign to social information, relative to their own individual estimate. Correspondingly, the predicted second estimate for an individual can be written as $E_2 = (1 - S_i) \cdot E_1 + S_i \cdot X$. When calculating the values of $S$, we omitted the rarely observed, cases in which participants moved away from social information ($s < 0$; 2.2% of cases) or moved beyond the social information ($s > 1$; 5.6% of cases). These represent qualitatively different cases in that the second estimate is not a weighted average of one's own individual estimate and social information [61]. The main results reported in this paper did not change when these cases were included in the analyses, or when these cases were treated as the nearest value in the range [0,1].

Participants did not receive feedback about their performance during the task. This prevented participants from learning about their own accuracy or the accuracy (or reliability) of the social information. After completing both blocks of 5 rounds, one of their estimates was randomly selected to count towards their earnings. When this estimate was exactly correct, a participant received 100 points. For each animal a participant was off the true value, 5 points were subtracted. Earnings could not fall below 0 points. So, participants were incentivised to only adjust their first estimate if they thought it would improve their accuracy. This procedure ensured a high level of experimental control without deception. Debriefing after pilot sessions confirmed that students did not doubt that the displayed estimates stemmed from actual people, and that they deemed social information potentially useful to improve the accuracy of their estimates.

At the end of the session, participants were compensated according to performance with stationary items, such as pencils and note books. Participants received two items for participation; for every 20 points they earned during the task, they could earn one additional (bonus) item. On average, participants earned 50 points (s.d. 34), which converted into an average of 2.21 (s.d. 1.64) bonus items. The experiment was programmed in *LIONESS Lab* [68]; experimental code is available upon request from the first author (L.M.).

## Analyses

Our main analyses focus on how model age (peer or adult) and household composition impacted participants' adjustments of estimates after observing social information. Our regression models included 'age' and 'gender' as control variables, and used 'participant nested in school' as random effects (exact specification of these models are detailed where they are presented). Analyses were conducted in *R* v. 3.5.1 [69]; for regressions we used the package 'lme4' [70]. For assumption checks for reported tests, see Supporting Text in S1 File. Our primary analyses use 'extended households' defined as grandparents being in the household or not,

irrespective of cohabiting aunts or uncles (see Table C in S1 File for sample composition with respect to cohabiting grandparents and aunts or uncles). The main results reported below are robust to defining extended households as in- or excluding aunts and uncles.

## Results

### Basic behavioural results

Participants' first estimates tended to be slightly lower than the true value (Panel A of Fig A in S1 File). Underestimation is a typical result in estimation tasks [42,62] and the magnitude of underestimation in this task (about 10% on average) was very similar to previous observations from adult participants recruited from Amazon Mechanical Turk [61], Colombian fishermen and farmers, Dutch university students, as well as British and German teenagers (manuscripts on these datasets are in preparation). In line with these other samples, second estimates were, on average, closer to the correct value (Panel B of Fig A in S1 File; linear mixed model comparing deviations from the true value between first and second estimates, with period nested in participant nested in school as random effect: $\beta$ = -1.884 (0.103), $t$ = -18.32, $P$<0.001), which is perhaps unsurprising because by design, social information pointed in the direction of the correct value in most of the cases (Methods). These aggregate results indicate that, overall, participants used social information to update their estimates and to improve the accuracy of their decisions.

### The impact of model age and household composition on social information use

In line with our predictions, estimates provided by adults had a stronger impact on adolescents' own estimates than those provided by peers (Fig 2). After observing an adult, average adjustments were 47.5% from the own first estimate towards the social information. After observing a peer, average adjustments were 43.7%, which is significantly less (paired t-test comparing distributions of $S_{peer}$ and $S_{adult}$: t = -2.192, d.f. = 262, $P$ = 0.029; Cohen's $d$ = 0.135). This result is corroborated by a (more principled) analysis based on a regression model indicating that average adjustments were significantly influenced by model type (main effect: $P$ = 0.021) but not by participants' age or gender (Table 1, Model 1).

In contrast to our predictions, we did not observe a systematic effect of household composition on the relative impact of peers versus adults on adolescents' social information use (Fig 2; compare the differences between the yellow and blue bars for nuclear versus extended households). This result is confirmed by the regression model (Table 1, Model 1), which does not detect a significant interaction between 'model type' and 'household type'.

### Adjustments in individual rounds

To get more insight into how participants used social information to adjust their estimates, we turn to behaviour in individual rounds. Fig 3 shows that the most common adjustments can be classified into three qualitatively different categories (or 'adjustment heuristics'; [64,67,71]. Pooling adjustments in both experimental conditions, we observe that participants frequently chose to not adjust at all ($s$ = 0, 'stay'; 27% of the rounds), copy the model's estimate ($s$ = 1, 'copy'; 18%), or compromise between their own first estimate and the model's estimate (taking some weighted average between the two estimates; $0 < s < 1$, 'compromise'; 51%). Tendencies to 'stay' or 'copy' did not differ across household compositions (Table 1, Models 2 and 3).

Fig 3 further suggests that participants were more likely to stay with their first estimate when social information was provided by a peer, whereas participants were more likely to

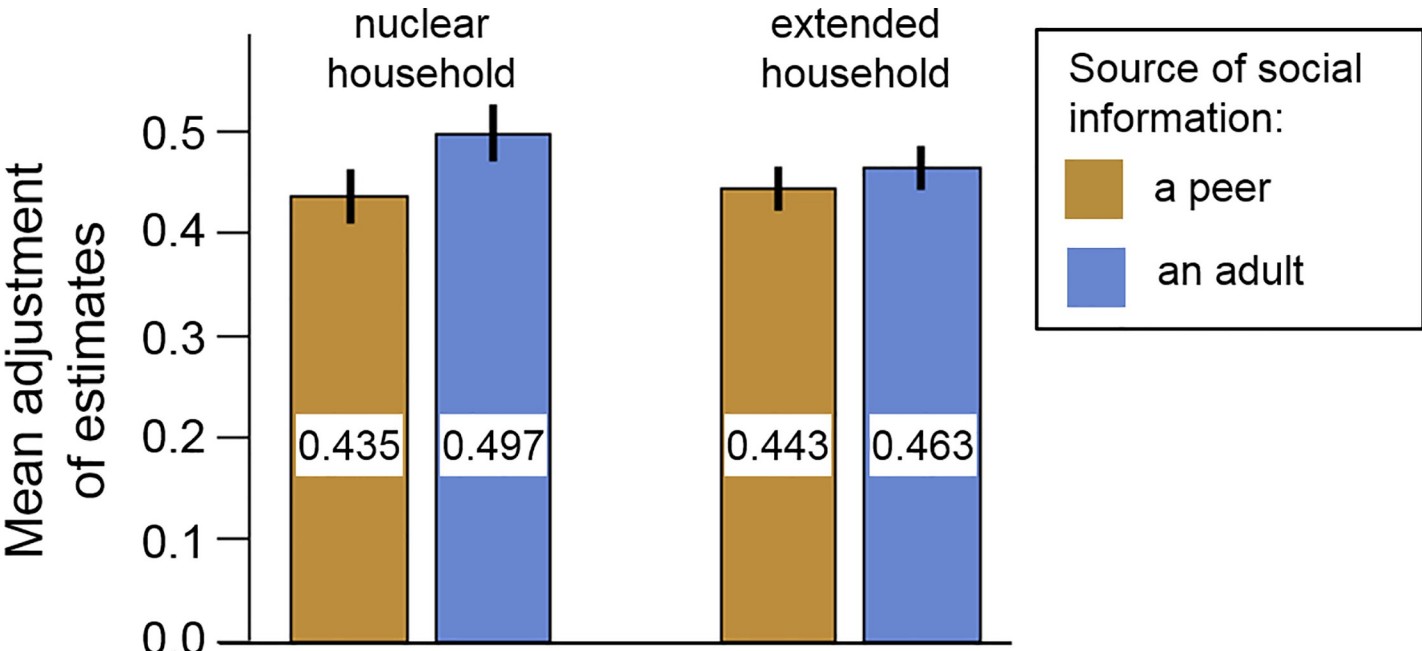

**Fig 2. Social information use depends on model age, not household composition.** Bars show mean adjustments in both experimental conditions (+/- 1 SEM), broken down by household composition of the participant and by model type. We observed that participants tended to adjust more when the source of social information was an adult. Household composition did not matter for this increase. For statistical analyses, see Table 1.

copy social information when it was provided by an adult. These observations are supported by logistic generalized linear models fitted to decisions to stay with one's own first estimate and decisions to copy the social information (main effect of 'model type' in models 2 and 3 of Table 1; $P = 0.008$ and $P < 0.001$, respectively). Note that these effects did not vary with household composition, but the significantly negative effect of 'age' in model 3 suggests that older adolescents were less likely to simply copy the estimates of other people.

**Table 1. Determinants of social information use.** The predictor 'model type' was coded as 0 when the source of social information was a peer and as 1 when the source was an adult. The predictor 'household type' was coded as 0 if a participant lived in a 'nuclear' household, and as 1 when they lived in an 'extended' household including grandparent(s). Gender was coded as 0 for males and 1 for females. Model 1 is a linear mixed model fitted to participants' mean adjustment ($S$) across the five rounds in an experimental condition. As dependent variable, this model included two data points for each participant: (i) mean adjustment when observing a peer ($S_{peer}$) and (ii) mean adjustment when observing an adult ($S_{adult}$). Model 2 was a logistic generalized mixed model (GLMM) fitted to decisions to *stay* with one's own first estimate (coded as 1) or move towards the social information (coded as 0). Model 3 was a logistic GLMM fitted to decisions to *copy* the social information (coded as 1 if social information was copied, 0 if it was not). In all models, we used 'participant nested in school' as random effect.

| | Model 1 | | Model 2 | | Model 3 | |
|---|---|---|---|---|---|---|
| | mean adjustment | | probability of 'stay' | | probability of 'copy' | |
| | estimate (SE) | *P* | estimate (SE) | *P* | estimate (SE) | *P* |
| **intercept** | 0.693 (0.236) | 0.004 | -2.023 (1.801) | 0.261 | 2.260 (2.445) | 0.355 |
| **model type** | 0.062 (0.027) | 0.021 | -0.480 (0.180) | 0.008 | 0.790 (0.211) | <0.001 |
| **household type** | -0.001 (0.035) | 0.971 | -0.139 (0.264) | 0.599 | -0.003 (0.359) | 0.993 |
| **model type x household type** | -0.040 (0.034) | 0.238 | 0.359 (0.228) | 0.115 | -0.477 (0.266) | 0.074 |
| **gender** | 0.018 (0.029) | 0.537 | 0.160 (0.235) | 0.497 | -0.126 (0.312) | 0.685 |
| **age** | -0.021 (0.017) | 0.204 | 0.050 (0.134) | 0.709 | -0.388 (0.180) | 0.031 |
| ***n*** | 511 | | 2,358 | | 2,358 | |
| ***N*** | 256 | | 256 | | 256 | |

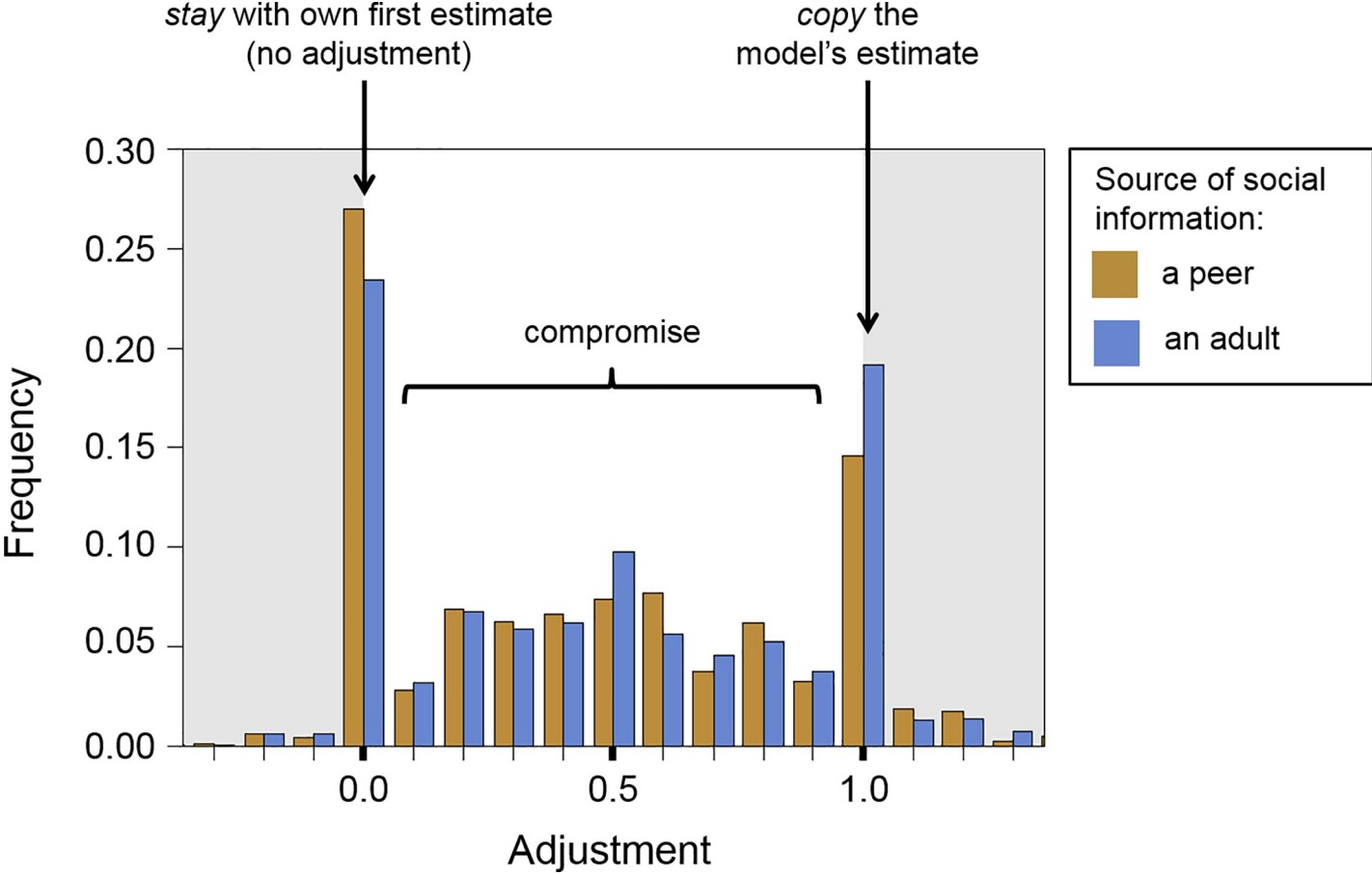

**Fig 3. Distribution of adjustments in individual rounds of the task, broken down by model age.** For each round, we calculate a participant's relative adjustment, that is, the fraction they moved toward the social information (see main text for details).

## Discussion

Our results indicate that model age but not household composition affected social information use in adolescents aged 11 to 15 years in urban India. On average, social information tended to have more impact on adolescents' behaviour when its source was an adult rather than a peer. However, this effect did not depend on household composition; the relative impact of adults was not more pronounced in adolescents from extended households than in those from nuclear households.

With average adjustments of 43% (when observing a peer) and 47% (when observing an adult), the level of social information use in our experiment was substantially higher than reported in studies of advice taking and social influence in adults [42,61,62,71–73]. These studies were largely conducted with adult participants from WEIRD (Western, Educated, Industrialized, Rich, and Democratic) societies [74] and found average adjustment levels of around 30–35%, indicating that people weight their own estimate about twice as much as the estimate of another individual. Although social information use in those studies was measured with different tasks, our data suggests that adolescents in our sample had lower levels of so-called 'egocentric discounting' [62]. Note that we cannot conclusively say whether these results are due to adolescents' increased sensitivity to social influence [34,35] or to the socio-cultural context of our study [75], or both. Due to its simplicity, minimal social context and easy implementation,

the paradigm used in our study might form a sound basis for systematic comparisons of the development of social learning across societies (in the spirit of, *e.g.*, [46]), disentangling the effects of age and socio-cultural environment. For useful discussions of some of the challenges of using cognitive tasks across cultures, see [76–78].

In contrast with the popular idea that adolescents only focus on their peers, we found that their behaviour was more influenced by adults than by peers. Our decision-making task measured sensitivity for social influence in a setting in which there was an objectively correct response (that is, the number of animals in an image). In that respect our results are in line with findings that adolescents were more influenced by adults than by peers when judging the probability of an event [37]. By contrast, peers have been found to have a greater impact on decisions when they involve social values or norms such as how to share resources with others [36]. Already from a young age, people are able to flexibly use social information depending on the source's past reliability or familiarity with the subject matter [79,80]. It is likely that adolescents in our study expected adults to be more skilled and to provide more accurate estimates than peer models. Thus, even in a period that peers are often considered the most important reference group, information of adults may still be valued. These findings underline the importance of studying the development of social information use in different domains (e.g., risk taking, inclinations for rule-following and compliance to social norms).

Our results provide a deeper insight into social information use in early to middle adolescence and how this may depend on characteristics of the model (*i.e.*, the source of social information). We found that adolescents in our Indian sample copied social information rather frequently (Fig 3), and did so more often when the model was an adult. Conversely, adolescents were more likely to ignore social information (*i.e.*, staying with their own first estimates) when the model was a peer. Previous studies with adults (mostly university students from Western countries) have reported occasional one-to-one copying of others' estimates, attitudes or opinions, but at much lower frequencies [42,62,71–73]. It is possible that the high levels of copying are specific to socio-cultural context of our study. Our observation that copying tends to become less frequent with age (Table 1, model 3) further suggests that the relative frequencies of simple adjustment heuristics decreases across development, in favour of an–arguably more complex–integrative compromising strategy. Future research could examine the extent to which our results generalize to different decision problems (see, for example [81–83]) and other (WEIRD and non-WEIRD) populations. Furthermore, the relative frequencies of adjustment heuristics (cf. 'stay', 'compromise' or 'copy' [71]) can have marked consequences for social dynamics at the collective level. For example, when people in a social network are repeatedly influenced by each other, these heuristics can give rise to distinct patterns of group decision-making such as the formation of consensus or polarization [64,67,84–87]. The relatively high frequencies of the 'extreme' heuristics of copying or staying (rather than compromising) suggest that, relative to adults, groups of adolescents may take longer to form a consensus and may be more likely to become internally polarized. Future theoretical and experimental studies could explore the possible implications of adolescents' social information use for longer-term dynamics of group decision-making.

Our study aimed to compare groups that differed in the variable of interest (household composition), but were otherwise very similar. Our analysis of adolescents' socio-demographic background confirms that the two groups (nuclear vs. extended family) were similar with respect to an–admittedly limited–set of measured control variables (Tables A and B in S1 File). Factors such as gender, age, socio-economic and societal background can have an impact on aspects of social cognition, including social learning [41,43,45,88,89] but it is often difficult to disentangle the effects of these factors in broader cross-cultural comparisons. Our approach allows for a more focused analysis of a single factor that might shape adolescents' social

information use, thereby complementing studies that make broader comparisons across cultures and across development [36,46]. The lack of differences in social information use between nuclear and extended households observed in our study may be explained by the fact that the transition from extended to nuclear households occurred relatively recently [49,59,60]. Although our data does not include measurements of family values or the history of their household composition, it is conceivable that people living in nuclear households may still uphold "extended" family values and receive social support from members of their extended family (*e.g.*, grandparents, uncles, aunts), who tend to live nearby.

Our study provides important new insights into the social information use of adolescents. We observed that adolescents showed high levels of social information use, especially when social information was provided by (presumably more knowledgeable) adults. These results highlight that there are situations in which adolescents put more weight on input from adults, instead of mostly focusing on peers. In addition, our data suggest that adolescents–or at least, Indian adolescents–frequently rely on simple adjustment heuristics ('stay' or 'copy') instead of using an integrative compromising strategy. Such developmental, or possibly cultural, differences in social learning strategies may have a significant impact on group behaviour. Finally, the current study provides a robust template to further investigate the developmental dynamics of social information use. Theoretically important aspects to study include the characteristics that may be important for selecting a model to learn from (*e.g.*, their expertise or popularity), how to integrate information from multiple sources (*e.g.*, when the influence of parents and peers operates in opposing directions: 'parents say no, peers say yes'), how to decide which social learning strategy fits best to the current situation, and how social information use is modulated by individuals' confidence in their own judgment [90,91]. Longitudinal and cross-sectional studies systematically tracking these aspects—mapping out their onset and development—could provide critical advances for understanding social learning and its determinants.

## Supporting information

**S1 File. This document contains Supporting Fig A, Supporting Tables A-C, Supporting Text, as well as the full experimental materials as shown to participants.**
(PDF)

## Acknowledgments

We are grateful to Dr. Jahnavi Sunderarajan for providing us with useful comments and discussions, as well as for her help collecting the data. We further thank Damien Fleur, Andrea Gradassi, Casper Hesp, Joshua Niebaum, Ana Pinho, Scarlett Slagter and Sjanne van der Stappen for useful comments and discussions.

## Author Contributions

**Conceptualization:** Lucas Molleman, Patricia Kanngiesser, Wouter van den Bos.

**Data curation:** Lucas Molleman.

**Formal analysis:** Lucas Molleman.

**Funding acquisition:** Lucas Molleman, Wouter van den Bos.

**Investigation:** Lucas Molleman, Patricia Kanngiesser, Wouter van den Bos.

**Methodology:** Lucas Molleman, Patricia Kanngiesser, Wouter van den Bos.

**Project administration:** Lucas Molleman.

**Resources:** Lucas Molleman.

**Software:** Lucas Molleman, Wouter van den Bos.

**Supervision:** Lucas Molleman, Patricia Kanngiesser, Wouter van den Bos.

**Validation:** Lucas Molleman, Patricia Kanngiesser, Wouter van den Bos.

**Visualization:** Lucas Molleman.

**Writing – original draft:** Lucas Molleman, Patricia Kanngiesser, Wouter van den Bos.

**Writing – review & editing:** Lucas Molleman, Patricia Kanngiesser, Wouter van den Bos.

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
