## [Decision Letter · Decision Letter 0]

15 Oct 2019

PONE-D-19-24671

Social information use in adolescents: the impact of adults, peers and household composition

PLOS ONE

Dear Dr. Molleman,

Thank you for submitting your manuscript to PLOS ONE. I sent it to a leading expert in this area of research and the feedback I received appears below. A number of positive aspects to your paper are outlined, but there are also multiple areas identified that demand attention. Of the latter, there is a need to clarify if the task you use captures social information use in the same way across cultures, detail about the language used in task administration, and if there was any piloting conducted to assess contextual validity. The review is clear and detailed so I will not elaborate further, but needless to say each point made deserves attention. After careful consideration, we feel that your manuscript has merit but does not fully meet PLOS ONE’s publication criteria as it currently stands. Therefore, we invite you to submit a revised version of the manuscript that addresses the points raised during the review process.

We would appreciate receiving your revised manuscript by Nov 29 2019 11:59PM. To enhance the reproducibility of your results, we recommend that if applicable you deposit your laboratory protocols in protocols.io, where a protocol can be assigned its own identifier (DOI) such that it can be cited independently in the future. For instructions see: http://journals.plos.org/plosone/s/submission-guidelines#loc-laboratory-protocols

A rebuttal letter that responds to each point raised by the academic editor and reviewer. This letter should be uploaded as separate file and labeled 'Response to Reviewers'.A marked-up copy of your manuscript that highlights changes made to the original version. This file should be uploaded as separate file and labeled 'Revised Manuscript with Track Changes'.An unmarked version of your revised paper without tracked changes. This file should be uploaded as separate file and labeled 'Manuscript'.

We look forward to receiving your revised manuscript.

Kind regards,

Mark Nielsen, Ph.D.

Academic Editor

PLOS ONE

Journal Requirements:

3. During our internal checks, the in-house editorial staff noted that you conducted research or obtained samples in another country. Please check the relevant national regulations and laws applying to foreign researchers and state whether you obtained the required permits and approvals. Please address this in your ethics statement in both the manuscript and submission information.

Reviewers' comments:

Reviewer's Responses to Questions

**Comments to the Author**

1. Is the manuscript technically sound, and do the data support the conclusions?

Reviewer #1: Yes

2. Has the statistical analysis been performed appropriately and rigorously? 

Reviewer #1: Yes

3. Have the authors made all data underlying the findings in their manuscript fully available?

Reviewer #1: Yes

4. Is the manuscript presented in an intelligible fashion and written in standard English?

Reviewer #1: Yes

5. Review Comments to the Author

Reviewer #1: This study examined adolescences’ whether social information use is influenced by the age of the model (peer vs adult) and/or participants’ household demographics (nuclear vs extended family). Using a number estimation task, results indicated that adolescents showed high levels of social information use, particularly when the source was an adult, while household composition had no effect on social information rates. There was also evidence of heuristics whereby adolescents demonstrated high rates of directly copying others estimates.

Overall, I really liked this study. In general, I had very few criticisms of the manuscript – I thought it was well written and I was very pleased to see that the participant sample was from India (and I really appreciated the use of sample-specific references). This is an important step for the cultural evolution literature to expand the range of populations tested beyond Western ones, and not just for comparisons. I was also extremely pleased that the manuscript specifically addressed adolescents. I personally advocate the need for our field to move beyond the young children/adult dichotomy that we seem to be stuck upon. As such, the question and findings are interesting and are clearly presented.

One thing I was interested in was whether the authors validated the task. A recent, growing body of cross-cultural studies have shown that tasks designed for use on Western populations often capture something different when taken to non-western populations. I acknowledge and understand that the BEAST was successfully implemented on a diverse population (from MTURK) – which is excellent and a good indication is translates well. But have the authors shown that the task is capturing social information use in the same way across cultures? What language was the task administered in (the screenshots suggest English, but I wasn’t sure if this was the version administered in this study)? If not English, what was the translation process (i.e. back translations)? Was there any piloting conducted to assess contextual validity (I note that there was some to capture appropriate timings, so perhaps a little more information here would be good)? This task strikes me as one that will translate well across cultures (especially urban populations in India and especially since accuracy results are similar to the MTurk findings), and so I’m not overly concerned about these questions, but they are ones that came to mind as I read the manuscript. If the authors intend to use this task to more non-Western populations (especially non-urban ones), I recommend Holding et al. (2018), Hruschka et al. (2018) and Zuilkowski et al. (2016) as extremely useful sources discussing applying simple cognitive tasks across cultures.

Minor comments:

Line 38: The authors might want to consider including Wood et al. (2013) here.

Line 47: I’m not sure the references 27-29 are appropriate here, since I don’t think these studies gave children the choice of who they preferred to copy from. More appropriate references would be Price et al. (2017) and Wood et al. (2013) and references therein. However, the authors should also see Wood et al. (2016) for evidence directly contrary to their statement.

Lines 86-88: I understand the sentient here, but as I’m certain the authors know India is an extremely diverse country (I have lived there!) and so I would rephrase this to be along the lines of ‘in some/most parts of India…’

The authors found that older adolescents were less likely to simply copy the estimates of others than younger adolescents. This is interesting, and somewhat counter to some overimitation work (e.g., McGuigan et al. 2011; Whiten et al. 2016), although it does fit with developmental work showing that older children are less likely to copy others than younger children (Carr et al, 2015). Do the authors have any ideas of why this might be?

Lines 303-308: Interesting – but most of these studies also used different tasks, which is likely to impact social information use (as the authors note in the introduction).

Methods: The age range is 11-15. This strikes me as early/middle adolescence rather than adolescence in general.

Method/results: I wondered (and couldn’t find the answer anywhere) whether extended households were comprised of just children, parents and grandparents, or whether extended households included other members (aunts, uncles, cousins, etc.). If the former, then I wondered whether the authors had any thoughts on whether including ‘more extended’ households including other family members than grandparents would influence social information use? If the latter, did the authors examine whether number of extended household members had any effect on social information use?

Results: Although not significant, the results suggest that participants from nuclear households were actually more likely to information from an adult (compared to a peer) than those from extended households. Do the authors have any thoughts on that?

Results: Were there any diagnostics/assumption meeting conducted on the data?

Discussion: As noted above, I really applaud the assessment of non-traditional experimental psychology populations (i.e. not WEIRD). However, as should be done in in those studies, the authors should consider mentioning/discussing that these findings should be considered within the context of the population studied. For instance, the authors state that “Our results provide a deeper understanding into adolescents’ social information use and how.…” (lines 330-333). The authors might want to highlight that these findings need to be verified across other populations (unless this specific study is replicated across populations). They do this very briefly in the final paragraph (which is great, but I would like to see it a little more in the main discussion).

Lines 338-340: Additionally, adolescence brings the maturation of cognitive mechanisms including executive functions. It’s possible that the decrease in copying with age is in part a result of better equipped cognitive profiles, and thus a lower need to rely on others for information for this sort of task.

Lines 341-350: I wonder whether the high frequency of extreme heuristics is a result of the quick nature of the task. That is, perhaps in a task with more time to develop strategies you might see more complex and more decision-making approaches.

Lines 363-365: Do the authors have any information of participant family values? If participants from nuclear families show similar such values to those in extended families (potentially indicating a recent shift), this argument would be presumably be supported.

An interesting addition to this study would be to see if individual confidence predicted the adjustment to social information (and in relation to adult/peer nuclear extended). Other studies with relatively similar paradigms have shown that task confidence is a predictor of social information use (Morgan et al. 2012; Cross et al. 2017).

References (in no particular order)

Wood, L. A., Kendal, R. L. and Flynn, E. (2013) ‘Whom do children copy? Model-based biases in social learning’, Developmental Review, 33(4), pp. 341–356. doi: 10.1016/j.dr.2013.08.002.

Price, E., Wood, L. A. and Whiten, A. (2017) ‘Adaptive cultural transmission biases in children and nonhuman primates’, Infant Behavior and Development. Elsevier Inc., 48, pp. 45–53. doi: 10.1016/j.infbeh.2016.11.003.

Wood, L. A. et al. (2016) ‘“Model age-based” and “copy when uncertain” biases in children’s social learning of a novel task’, Journal of Experimental Child Psychology, 150, pp. 272–284. doi: 10.1016/j.jecp.2016.06.005.

Holding, P. et al. (2018) ‘Can we measure cognitive constructs consistently within and across cultures? Evidence from a test battery in Bangladesh, Ghana, and Tanzania’, Applied Neuropsychology: Child. Psychology Press, 7(1), pp. 1–13. doi: 10.1080/21622965.2016.1206823.

Zuilkowski, S. S. et al. (2016) ‘Dimensionality and the Development of Cognitive Assessments for Children in Sub-Saharan Africa’, Journal of Cross-Cultural Psychology. SAGE PublicationsSage CA: Los Angeles, CA, 47(3), pp. 341–354. doi: 10.1177/0022022115624155.

Hruschka, D. J. et al. (2018) ‘Learning from failures of protocol in cross-cultural research.’, Proceedings of the National Academy of Sciences of the United States of America. National Academy of Sciences, 115(45), pp. 11428–11434. doi: 10.1073/pnas.1721166115.

Morgan, T. J. H. et al. (2012) ‘The evolutionary basis of human social learning’, Proceedings of the Royal Society B: Biological Sciences, 279(1729), pp. 653–662. doi: 10.1098/rspb.2011.1172.

Cross, C. P. et al. (2017) ‘Sex differences in confidence influence patterns of conformity’, British Journal of Psychology, 108(4), pp. 655–667. doi: 10.1111/bjop.12232.

Whiten, A. et al. (2016) ‘Social learning in the real-world: “over-imitation” occurs in both children and adults unaware of participation in an experiment and independently of social interaction’, PLOS ONE. Edited by R. L. Kendal. Public Library of Science, 11(7), p. e0159920. doi: 10.1371/journal.pone.0159920.

McGuigan, N., Makinson, J. and Whiten, A. (2011) ‘From over-imitation to super-copying: Adults imitate causally irrelevant aspects of tool use with higher fidelity than young children’, British Journal of Psychology, 102(1), pp. 1–18. doi: 10.1348/000712610X493115.

Carr, K., Kendal, R. L. and Flynn, E. (2015) ‘Imitate or innovate? Children’s innovation is influenced by the efficacy of observed behaviour’, Cognition, 142, pp. 322–332. doi: 10.1016/j.cognition.2015.05.005.

6. PLOS authors have the option to publish the peer review history of their article (what does this mean?). If published, this will include your full peer review and any attached files.

Reviewer #1: Yes: Bruce Rawlings

---

## [Author Response · Author response to Decision Letter 0]

5 Nov 2019

For our response to the reviewer and editor comments, please see the separate Response to Reviewers document.

---

## [Editor Report · Decision Letter 1]

7 Nov 2019

Social information use in adolescents: the impact of adults, peers and household composition

PONE-D-19-24671R1

Dear Dr. Molleman,

Thank you for your careful and considered revision. I am satisfied with your responses and associated changes in the manuscript. I am thus pleased to inform you that your manuscript has been judged scientifically suitable for publication and will be formally accepted for publication once it complies with all outstanding technical requirements.

With kind regards,

Mark Nielsen, Ph.D.

Academic Editor

PLOS ONE

---

## [Editor Report · Acceptance letter]

15 Nov 2019

PONE-D-19-24671R1 

Social information use in adolescents: the impact of adults, peers and household composition 

Dear Dr. Molleman:

I am pleased to inform you that your manuscript has been deemed suitable for publication in PLOS ONE. Congratulations! Your manuscript is now with our production department. 

With kind regards,

on behalf of

Dr. Mark Nielsen 

Academic Editor

PLOS ONE